# Transcriptional Profiling of Whisker Follicles and of the Striatum in Methamphetamine Self-Administered Rats

**DOI:** 10.3390/ijms21228856

**Published:** 2020-11-23

**Authors:** Won-Jun Jang, Taekwon Son, Sang-Hoon Song, In Soo Ryu, Sooyeun Lee, Chul-Ho Jeong

**Affiliations:** 1College of Pharmacy, Keimyung University, Daegu 42601, Korea; mrdoin76@gmail.com (W.-J.J.); gnsdl0330@naver.com (S.-H.S.); 2Research Institute of Pharmaceutical Sciences, College of Pharmacy, Seoul National University, Seoul 08826, Korea; taekwon@gmail.com; 3Substance Abuse Pharmacology Group, Korea Institute of Toxicology, Daejeon 34114, Korea; insoo.ryu@kitox.re.kr

**Keywords:** methamphetamine, drug reward, RNA sequencing, drug addiction

## Abstract

Methamphetamine (MA) use disorder is a chronic neuropsychiatric disease characterized by recurrent binge episodes, intervals of abstinence, and relapses to MA use. Therefore, identification of the key genes and pathways involved is important for improving the diagnosis and treatment of this disorder. In this study, high-throughput RNA sequencing was performed to find the key genes and examine the comparability of gene expression between whisker follicles and the striatum of rats following MA self-administration. A total of 253 and 87 differentially expressed genes (DEGs) were identified in whisker follicles and the striatum, respectively. Multivariate and network analyses were performed on these DEGs to find hub genes and key pathways within the constructed network. A total of 129 and 49 genes were finally selected from the DEG sets of whisker follicles and of the striatum. Statistically significant DEGs were found to belong to the classes of genes involved in nicotine addiction, cocaine addiction, and amphetamine addiction in the striatum as well as in Parkinson’s, Huntington’s, and Alzheimer’s diseases in whisker follicles. Of note, several genes and pathways including retrograde endocannabinoid signaling and the synaptic vesicle cycle pathway were common between the two tissues. Therefore, this study provides the first data on gene expression levels in whisker follicles and in the striatum in relation to MA reward and thereby may accelerate the research on the whisker follicle as an alternative source of biomarkers for the diagnosis of MA use disorder.

## 1. Introduction

Methamphetamine (MA) is a highly addictive psychostimulant whose abuse frequently leads to physical and psychological dependence. In recent years, there was a dramatic increase in the use of MA in North America [1] and in East Asian countries, including Korea [2]. Preclinical and clinical studies indicate that MA exposure results in substantial physical and psychiatric impairments such as depression, cognitive deficits, and psychotic behavior [3]. Repeated consumption of drugs such as cocaine and MA can cause the transition from casual to compulsive drug use, and this transition is thought to be initiated by neuroadaptive changes in brain circuits involved in drug rewards and cognitive processes that regulate habitual behaviors [4,5]. These neuroadaptive changes are commonly caused by complicated alterations of gene expression implicated in transcriptional regulation, synaptic plasticity, and intracellular signaling pathways in brain regions related to drug addiction [6,7]. Therefore, it may be worthwhile to investigate the change of gene expression patterns in the striatum during MA self-administration for speeding up the diagnosis and treatment of MA use disorder. 

There are major obstacles to the search for novel relevant biomarkers in the brain: primarily the difficulty with obtaining brain tissue from living donors and the lack of reliable experimental animal models. The brain is an ectodermal tissue and shares its developmental origin with scalp hair follicles, which are readily accessible mini-organs in the skin. Recently, Yoshikawa and colleagues proposed hair follicles as a beneficial genetic resource for the identification of potential biomarkers of autism and chronic psychosis (as in schizophrenia) [8]. After co-examining the gene expression in hair follicles and post-mortem brain tissue samples from patients with autism, they suggested that hair follicles may serve as an alternative apparatus reflecting the state of the central nervous system [8]. Consistently with this finding, our previous data have revealed that alterations of gene expression patterns in the whisker follicles of MA self-administered rats may serve as useful indicators of MA rewards [9]. However, to date, large-scale evaluation and comparison of the molecular changes have not been conducted between hair follicles and the striatum. 

High-throughput RNA sequencing (RNA-seq) is a high-productivity transcriptome profiling method designed to quantify changes in RNA expression, thereby revealing the functional output of a disease state such as a drug use disorder. To understand how to diagnose and treat MA addiction, it is crucial to find hub genes and to delineate their neurobiological functions related to MA rewards and addiction. Numerous relevant biological events point to the need for large-scale analyses to uncover the diverse pathways and mechanisms that are most prominently involved in MA addiction. In this study, a rat MA self-administration model [10] was used to study potential molecular phenomena underlying MA rewards, and the collected gene information was analyzed by means of several bioinformatics tools, such as multivariate analysis and network-based topological analysis.

This study provides data on overall gene expression in whisker follicles and in the striatum of the same rats following MA self-administration. Using genome-wide quantification of transcripts, we aimed to find and clarify the role of concordantly and differentially expressed genes (DEGs) in the two tissues and to reveal the gene–gene interaction mechanisms behind the rewarding effect of MA.

## 2. Results

### 2.1. Methamphetamine Self-Administration

This procedure was performed on rats according to the experimental schedule illustrated in Figure 1A. Figure 1B,C show the number of active and inactive lever responses during saline (SA) or MA self-administration (2 h/day, 16 days) under the fixed-rate 1 (FR1) schedule. The MA group showed clear goal-oriented behavior with a significantly higher number of active lever presses than that of the SA group (self-condition: *p* < 0.0001; subjects (matching): *p* < 0.001). No significant differences in the numbers of inactive lever presses were observed between both groups (self-condition: *p* = 0.3177; subjects (matching): *p* < 0.0001). The number of drug infusions was significantly higher in MA self-administering rats than in SA self-administering rats (self-condition: *p* < 0.0001; subjects (matching): *p* < 0.0001) and showed <10% variation during the last 3 days of the experiment (Figure 1D). The number of MA infusions on the final day was 23 ± 4.2 (mean ± SEM (standard error of mean)). Two-way repeated measure analysis of variance (ANOVA) uncovered a significant difference in the number of infusions between SA and MA self-administered rats. In addition, we assessed the expression of several marker proteins in the dopamine signaling pathway in the striatum of rats exposed to MA self-administration. Our data revealed that the expression levels of a D1 dopamine receptor (D1DR) and brain-derived neurotropic factor (BDNF) that play a role in the regulation of synaptic plasticity were significantly increased in the striatum of the methamphetamine self-administered rats compared with those of the saline control. In addition, the phosphorylation of tyrosine hydroxylase (p-TH, ser40) was significantly increased, but the total level was not changed in the striatum of the methamphetamine self-administered rats (Appendix A). These results are consistent with previous reports indicating that these proteins could be regulated by exposure to MA or other drugs of abuse such as cocaine [11,12].

### 2.2. RNA-Seq Analysis and Gene Expression Profiling

Samples were sequenced on the Illumina HiSeq 2000 platform, with paired-end 101 bp reads for mRNA-seq using the TruSeq SBS Kit v3 (Illumina). The raw imaging data were transformed by base-calling into sequence data and stored in FASTQ format. After removing the adapters, approximately 39.22 million reads could be detected in each sample. Alignment results from TopHat2 showed that the unique mapping rate was 91.52% on average, and 17,711 genes could be found when compared against the rat reference genome rn6. Integrative RNA-seq profiling was performed on whisker follicle samples and striatum samples collected from six SA self-administered and six MA self-administered rats. A total of 87 genes (false discovery rate (FDR) <0.05, |fold change (FC)| ≥ 1.5) were found to be differentially expressed in the striatum in the MA group relative to the SA group (Appendix A), and 253 DEGs with FDR< 0.05 and |FC| ≥ 1.5 were identified in whisker follicles (Appendix A). MA self-administered rats showed a significant down-regulation of 14 genes in the striatum and 182 genes in the whisker follicle, whereas 73 genes in the striatum and 71 genes in the whisker follicle were significantly up-regulated as compared with SA self-administered rats. The hierarchical clustering revealed that the identified genes were highly consistent between two groups of each whisker follicle and the striatum tissue (Appendix A). Next, we conducted a series of bioinformatics analyses to study the RNA-seq data according to the procedure presented in Figure 2. We analyzed the enriched function and pathway of up-regulated and down-regulated DEGs in the striatum (Figure 3A–D) and the whisker follicle (Figure 3E–H), respectively, using the Database for Annotation, Visualization, and Integrated Discovery (DAVID) online tool based on the two complementary databases of Gene Ontology (GO) and the Kyoto Encyclopedia of Genes and Genomes (KEGG). The *p*-value and gene count were set to <0.05 and ≥2 as significant thresholds for enrichment analysis in DAVID tool.

### 2.3. Multivariate Analysis

We analyzed 87 DEGs in the striatum and 253 DEGs in the whisker follicle from the RNA-seq data on the two tissues. To check the quality of the data and evaluate the magnitude of separation from the control, principal component analysis (PCA) was applied to the DEGs from the SA and MA groups of the two tissues. Expectedly, the PCA score plot of DEGs showed a much more clear separation between two groups in each tissue than that of total genes (Appendix A). As shown in Appendix A, 2D score scatter plots of PCA mostly occupied the space of principal component 1 (PC1) and principal component 2 (PC2). In the striatum, contributions of PC1 and PC2 to the maximum variance in the dataset were calculated and found to be 48.1% and 20.5%, respectively (Appendix A). In addition, PC1 and PC2 in the whisker follicle covered 81.9% and 4.5% of the variance in the dataset, respectively (Appendix A). In both tissues, most samples in the MA groups were clearly separated from the SA groups (Appendix A). Next, to identify the critical genes that contribute to the separation between the groups, we performed correlation analysis and correlation pattern searching (Appendix A). Pearson’s correlation coefficients for each gene were calculated. To screen out the critical genes, the correlation coefficient for each gene was limited to 0.6 or higher (relative coefficient *r* ≥ 0.6 is thought to denote a strong correlation). Thirty critical genes in the striatum (Appendix A) and 72 critical genes in the whisker follicle (Appendix A) were identified, and the top 25 were represented by means of a correlation pattern searching tool (Appendix A). The positive and negative correlations of the genes are indicated by the red and blue color, respectively.

### 2.4. Construction of Protein–Protein Interaction (PPI) Networks

Next, we constructed protein–protein interaction (PPI) networks from total, up-regulated, and down-regulated DEGs of the two tissues, respectively, based on the Search Tool for the Retrieval of Interacting Genes/Proteins (STRING) database [13] (Appendix A). The PPI networks were built using the Cytoscape software. The fold change of each node was expressed as color intensity. The up- and down-regulated networks of each tissue showed a separate form from the total network, but no networks were lost or newly formed. We analyzed the enriched function and pathway of the six networks using the DAVID tool and summarized the results in Appendix A with biological process (BP), cellular component (CC), molecular function (MF) functional tools as well as KEGG. The total network of the striatum consisted of 69 nodes and 72 edges and appeared to contain several fragmented modules (Figure 4A). Most of the genes that make up the striatum network were found to be up-regulated by MA. The total network of whisker follicle consisted of 223 nodes and 1191 edges and looked like a single network that was densely packed around the center (Figure 4B). Genes located near the center of the whisker follicle network tended to be down-regulated by MA, while some genes located outside the center were found to be up-regulated by MA

### 2.5. Cluster Analysis

We performed a cluster analysis on the PPI networks of the two tissues to elucidate the network structure and the relations among its components [14,15]. The cluster analysis was conducted using the Molecular Complex Detection (MCODE) plug-in of the Cytoscape software, which identifies highly interactive nodes with high scores in terms of the function and localization similarity of proteins [15]. Using the cluster analysis by MCODE, we found top three statistically significant gene modules in the network of the striatum and of the whisker follicle, respectively (Figure 4). To verify the biological functions of each module in the networks, we performed pathway enrichment functional analysis by means of the ClueGO application based on the KEGG database (Appendix A). Our data revealed that module 1 of the striatum network was significantly enriched with addiction-related functions such as nicotine addiction, cocaine addiction, amphetamine addiction, glutamatergic synapse, retrograde endocannabinoid signaling, and osteoclast differentiation signaling (Figure 5A). Notably, *FosB*, *Fosl2*, *Arc*, and *Junb* were identified as statistically significant genes that were up-regulated after MA self-administration, whereas *Gabra1*, *Slc17a6*, *Slc17a7*, and *Gria2* were identified as down-regulated genes (Figure 5A). Next, we performed pathway enrichment functional analysis on two representative modules of whisker follicles. As a result, functionally relevant pathways were analyzed in module 1 and module 2 (Figure 5B,C). Module 1 of whisker follicles turned out to be enriched with pathways associated with neurodegenerative diseases (such as Huntington’s disease, Alzheimer’s disease (AD), and Parkinson’s disease) and oxidative phosphorylation as well as a pathway related to ribosomes (Figure 5B). Of note, module 2 of whisker follicles was also mainly enriched with pathways associated with neurodegenerative diseases, although all genes of module 2 are different from those of module 1 (Figure 5C). Moreover, *App*, *Uchl1*, and *Ap2a1* were identified as statistically significant genes whose expression increased after MA self-administration, whereas all the genes that make up these two modules featured a down-regulation pattern. Overall, 15 genes in the striatum and 62 genes in the whisker follicle were selected by the cluster analysis.

### 2.6. Centrality Analysis

For the topological analysis of the network, the degrees and betweenness centrality values were calculated with the NetworkAnalyzer tool [15] of the Cytoscape software. To identify hub genes that are important for network cross-talk and biological essentiality [16,17], two topological features, the degree and betweenness centrality, were considered. In each network (the striatum and the whisker follicle), the degrees were set to cut-off levels of 2 and 5, respectively. The genes filtered by degree values are listed in the order of betweenness centrality values, and each of the top 20 genes was chosen as a hub gene. The hub genes of each tissue were visualized in scatter plots with the degrees and betweenness centrality values as axes (Figure 6A,B, left panel). A list of hub genes is shown at the right of Figure 6A,B. KEGG pathway analysis shown in Appendix A revealed that the hub genes of the striatum were mainly enriched with addiction-related functions and nervous system-related functions. The hub genes of the whisker follicle were enriched with neurological disorder-related functions.

### 2.7. Identification of Potential Diagnostic Markers

In the multivariate analysis and network analysis, 49 and 129 statistically significant genes were eventually selected from the DEGs of the striatum and of the whisker follicle, respectively, and these chosen genes are presented in a Venn diagram according to the statistical analysis performed (Figure 7). Of the 49 genes selected in the striatum, the activity-regulated cytoskeletal gene (*Arc*) and Jun B proto-oncogene (*Junb*) were found to be crucial genes selected by each of the three statistical analyses (Figure 7A). In the whisker follicle, the amyloid precursor protein (*App*) gene was also found to be a crucial gene selected by all three statistical analyses (Figure 7B). Notably, *Per1*, *Ddit4*, and *Tagln* were identified as common genes between whisker follicles and the striatum (Figure 7A,B). We also verified the expression of these key genes by Western blot and RT-PCR analyses (Appendix A). Next, to examine the comparability of gene expression between whisker follicles and the striatum, we conducted functional analysis of relations of the two tissues using the ClueGO application in the Cytoscape software. In Figure 8, tissue-specific functions in the striatum (blue) and in the whisker follicle (green) and common functions between the two tissues (yellow) are presented. Notably, several pathways, including synaptic vesicle cycle and retrograde endocannabinoid signaling pathways, were identified as common functions that link the whisker follicle tissues and striatal tissues, indicating the cross-talk between both tissues.

## 3. Discussion

Drug addiction is a complicated chronic disorder that is conceptualized as a cycle that includes a reward effect, withdrawal effect, craving, and a relapse to drug-seeking behavior [18,19]. The short-access self-administration model used in the present study is designed to study the reward process, which is an initial step of MA addiction. To find potential diagnostic genes in this pathological process, the transcriptional changes caused by MA self-administration were investigated in whisker follicles and in the striatum. Previously, we have explored the changes in gene expression in the whisker follicle of MA self-administered rats to identify novel indicators of the MA reward and found that gene expression in the whisker follicle could be used to understand the rewarding effect of MA in the striatum [9]. Nonetheless, nobody has directly examined the concomitant gene expression during MA self-administration both in the striatum and in whisker follicles of the same rat at the same time points, nor have any researchers placed their results in the context of systems biology. Therefore, the examination of differential gene expression in a discrete brain region, the striatum, and at an alternative sampling site (whisker follicles) will reveal an additional transcriptional landscape for MA rewards. Furthermore, it might be interesting to investigate whether the non-invasive analysis of whisker follicles is comparable to gene expression analysis in the striatum and yields similar results. Moreover, our study provides convergent systems biological evidence of genetic networks common between whisker follicles and the striatum in rats, and this evidence could further elucidate MA addiction in humans.

Multivariate analysis is a useful statistical technique for understanding the structure and relations of a dataset and for evaluating a biomarker candidate with respect to various experimental measurements [20,21]. As multivariate analyses, PCA and correlation analysis were conducted here using the MetaboAnalyst v4.0 (http://www.metaboanalyst.ca) online tool. Both PCA and correlation analysis can visualize variation in a dataset and identify technical or biological outliers that were not excluded at the alignment step. Topological analysis based on a PPI network plays an important role in the prediction of the function of genes or proteins connected to a network and provides a view into the functional relations of the interactions among the genes or proteins in a biological system [15,22]. PPI networks generally have a variety of topological features such as the degree, betweenness centrality, closeness centrality, shortest path length, and clustering coefficient that help to clarify the architecture and function of the network [15,23]. Although each of these analyses has its own advantages, there is also a risk that the data may be misinterpreted or missed if only one type of analysis is applied. Therefore, we employed multiple bioinformatics tools to analyze the RNA-seq data, and this strategy might be more reliable than a single-analysis approach in that the multiple analyses can reduce error and improve the authenticity of the data.

Independent multivariate and network analyses detected 49 and 129 genes (including *Arc*, *Junb*, and *App*, which are expected to play a major role in MA reward and addiction) in the striatum and whisker follicle, respectively. Some reports indicate that an acute injection of MA significantly up-regulates many immediate early genes such as members of the AP-1 family [24,25], and that acute and chronic administration of MA leads to increased *Arc* expression in the rat cerebral cortex, striatum, and hippocampus [26]. Our data revealed that several AP-1 family members such as *Junb*, *Fosl2*, and *Fosb* were up-regulated by MA self-administration in the striatum. The MA self-administered rats showed increased expression of a synaptic-plasticity-related gene, *Arc*, in the striatum. *Arc* is an effector immediate early gene participating in synaptic plasticity and memory consolidation [27,28]. An increased expression of *Arc* in the visual cortex has been reported to decrease synaptic strength by inducing α-amino-3-hydroxy-5-methyl-4-isoxazolepropionic acid (AMPA) receptor endocytosis [29,30], which is consistent with our data showing decreased expression of *Gria2*, which is a gene encoding a subunit of the AMPA receptor.

Our results revealed that highly connected gene modules are associated with drug addiction in the striatum. By contrast, in whisker follicles, DEGs known (through annotation) to be associated with neurodegenerative disorders were prominent. These findings imply that MA self-administration induces differential gene expression in both tissues, thus activating distinct functions. However, in this context, it is noteworthy that *Arc* is required for postsynaptic trafficking and processing of Appand amyloid β formation, which appear to be relevant to AD pathogenesis [31]. In whisker follicles, the *App* gene has been identified as a potential marker gene, which is in agreement with our previous data suggesting that *App* is an addiction-related gene in the whisker follicle in MA self-administered rats [9]. Although much remains to be studied, we cannot ignore the possibility that MA-induced genetic changes disturbing homeostatic synaptic plasticity in the brain are somehow reflected in whisker follicles, thereby linking the functions between striatal tissues and whisker follicle tissues. In fact, a recent report supports this hypothesis in that the pathological changes associated with AD can be reflected in peripheral tissues, fluids, and cells such as fibroblasts, the olfactory epithelium, whole blood, and plasma, which are considered useful sources for potential biomarkers of AD [32].

Moreover, in our study, several DEGs including the circadian clock gene period 1 (*Per1*) were identified in both the striatum and whisker follicles, i.e., in the overlap between their gene sets. Drug addiction has long been linked to disruptions in diurnal and circadian rhythms. Of note, it has been demonstrated that drugs of abuse alter the expression of circadian-clock genes in reward-related regions of the brain, thereby directly affecting ongoing circadian rhythms [33]. In addition, people with insomnia are more prone to addiction [34], suggesting that the disruption of circadian rhythms may be strongly related to drug reward and addiction. In agreement with this notion, acute MA administration leads to a rapid induction of *Per1* expression in the dorsal striatum of mice [35]; this change may increase drug seeking and craving when drugs are anticipated. Additionally, a recent report suggests that *Per1* expression in peripheral cells such as leukocytes and hair follicles may be used as an appropriate biomarker for the assessment of biological clock traits [36]. Therefore, our findings strongly indicate that *Per1* expression in peripheral hair follicles is an indicator of MA reward in the brain. 

Collectively, our findings provide overall gene expression profiles and identify key genes and pathways related to MA reward in whisker follicles and in the striatum as a result of a series of bioinformatics analyses. The gene expression patterns were found to be mostly different between whisker follicles and the brain, pointing to the activation of independent functions in the two tissues in response to the MA. The gene expression patterns were found to be mostly different between whisker follicles and the brain, pointing to the activation of independent functions in the two tissues in response to the MA. Nevertheless, our findings suggest that common alterations of expression of several genes and pathways in the brain and peripheral whisker follicles may serve as diagnostic markers of the rewarding effect of MA. Future research directions may also be highlighted.

## 4. Materials and Methods 

### 4.1. Animals

Male Sprague–Dawley rats (Daehan Animal, Seoul, Korea) weighing between 260 and 280 g at the start of the study were housed individually on a 12 h light-and-dark cycle, with all the experiments conducted during the light cycle. The rats had ad libitum access to food and water throughout the experiment. All the animals were handled (for acclimation) for at least 5 days prior to the surgical procedure and experiments. All experimental procedures were approved by the Institutional Animal Care and Use Committee at the Korea Institute of Toxicology (Approval No. KIT-1605-0710, B116061, 12-09-2016) and met the Association for Assessment and Accreditation of Laboratory Animal Care International guidelines for the care and use of laboratory animals.

### 4.2. Methamphetamine Self-Administration and Sample Collection

Before the MA self-administration experiment, rats were trained to press the active (drug-paired) lever (30 min/day for 3 days) for a food pellet reward. Only the rats that earned at least 100 pellets in the last session of the training were chosen for the further experiment. Surgical and post-surgical techniques have been described in detail in other studies [37,38]. After surgery, the rats were allowed to recover for 7 days. The recovered rats were moved to a special chamber, and either MA dissolved in SA or SA alone was infused into the rat when the lever was pressed. The MA self-administration was performed in daily 2 h sessions (0.05 mg/kg per intravenous infusion) for 16 days. The numbers of lever presses and infusions were analyzed by two-way repeated measure (RM) analysis of variance (ANOVA) with modality of self-administration (MA, SA) and days as between- and within-subject factors, respectively, using GraphPad Prism 8.0.2. After two-way RM ANOVA, Bonferroni multiple comparison tests were used for post-hoc analysis. After the end of the MA self-administration protocol, whisker follicles and the striatum were collected as described elsewhere [9,39]. The collected tissue samples were stored at −80 °C until analysis.

### 4.3. RNA Extraction

Total RNA was isolated from whisker follicles and the striatum by the conventional method (with the TRIzol^®^ reagent; Invitrogen, Carlsbad, CA, USA). The quality and concentration of RNA were assessed on a filter-based multi-mode microplate reader (FLUOstar Omega, BMG Labtech, Ortenberg, Germany). Residual genomic DNA was removed by incubation with RNase-free DNase I for 30 min at 37 °C.

### 4.4. Construction of Transcriptome Libraries

The isolated total RNA was processed for preparation of an mRNA sequencing library using a TruSeq RNA Sample Preparation Kit v2 (Illumina, San Diego, CA, USA) according to the manufacturer’s instructions. In brief, mRNAs were isolated from 1 μg of total RNA on RNA purification beads by polyA capture, which was followed by enzymatic shearing. After first- and second-strand cDNA synthesis, A-tailing and end repair were performed for the ligation of proprietary primers that incorporate unique sequencing adaptors with an index for tracking Illumina reads from multiplexed samples run on a single sequencing lane. For each library, an insert size of approximately 200-500 bp was confirmed by an Agilent 2100 Bioanalyzer (Agilent Technologies, Palo Alto, CA, USA), and quantification of the library was carried out by real-time PCR. Samples were sequenced on the Illumina HiSeq 2000 platform, with paired-end 101 bp reads for mRNA-seq using the TruSeq SBS Kit v3 (Illumina). The raw imaging data were transformed by base-calling into sequence data and stored in FASTQ format.

### 4.5. Processing of the RNA-Seq Data

Paired-end reads of the 24 independent samples were trimmed with Cutadapt to remove both PCR and sequencing adapters [40]. The trimmed reads were aligned to the rn6 rat reference genome in the TopHat2 software [41]. The quantification of gene expression was performed with Cufflinks [42] to calculate the fragments per kilobase of transcript per million mapped reads (FPKM) values.

### 4.6. Statistical Analysis of Differentially Expressed Genes (DEGs)

The presence of statistically significant differential expression was determined in Cuffdiff [43] at the gene level. Then, we chose gene expression differences based on their average FDRs and average log2-fold expression changes between the SA self-administration group (control) and the MA self-administration group in whisker follicles and in the striatum. For these comparisons, we assumed that genes are differentially expressed if the FDR was below 0.05 and the absolute value of fold change was >1.5.

### 4.7. Principal Component Analysis (PCA) and Hierarchical Clustering

These analyses were performed by means of MetaboAnalyst v4.0 (https://www.metaboanalyst.ca) [44]. The PCA was intended to discriminate between MA and SA samples of each tissue on the basis of their RNA-seq profiles. The values of DEGs were transformed by the autoscaling method (mean-centered and divided by the standard deviation of each variable) in MetaboAnalyst; 2D score plots of PCA between principal component 1 (PC1) and 2 (PC2) were constructed next. Red and green areas denote 95% confidence regions of MA and SA groups, respectively. The hierarchical clustering was also performed on MA and SA samples of each tissue. Euclidean distances were used as a distance measured between samples.

### 4.8. Correlation Analysis

Correlation matrix and pattern analyses of DEGs from the striatum and from the whisker follicle were performed using MetaboAnalyst. The correlation analysis was intended to evaluate the strength of a linear relation between different DEGs of each tissue. The raw data on DEGs were converted by the autoscaling method of the MetaboAnalyst online tool. For correlation matrix and pattern analyses, Pearson’s correlation coefficient (Pearson’s *r*) was computed as a correlation parameter (distance measure). Pearson’s *r* adopts a value between +1 and −1, where +1 is a total positive linear correlation, and −1 is a total negative linear correlation. The values of Pearson’s *r* for the entire gene set were obtained using PatternHunter in MetaboAnalyst, and the top 25 genes were visualized in a bar chart.

### 4.9. Centrality Analysis

PPI networks were constructed using the STRING application in Cytoscape (https://cytoscape.org). The DEGs of whisker follicles and of the striatum were employed for a protein query in the STRING application for analysis of the PPI networks. The confidence cut-offs and maximum additional interactors of the network visualization were set to 0.4 and 0.0, respectively. The color intensity of each node represented a fold change. To clarify the functional and biological importance of the genes, we analyzed topological structures of the PPI networks. For the determination of hub genes in the PPI networks of each tissue, the degrees and betweenness centrality values based on topological features were calculated via the NetworkAnalyzer tool in Cytoscape. The cut-offs for the degree in the networks of the striatum and of the whisker follicle were set to 2 and 5, respectively. The genes in the networks were ranked in the order of increasing betweenness centrality values, and the top 20 genes of the lists were designated as hub genes.

### 4.10. Functional and Pathway Enrichment Analyses

For functional and pathway enrichment analysis of the DEGs or PPI networks, two complementary analyses of Gene Ontology (GO, http://geneontology.org/), including biological process (BP), cellular component (CC) and molecular function (MF), and Kyoto Encylopedia of Genes and Genomes (KEGG, https://www.genome.jp/kegg/) were performed. The Database for Annotation, Visualization, and Integrated Discovery (DAVID, web-based online tool (https://david.ncifcrf.gov)) was used to perform GO functional and the KEGG pathway enrichment analysis. The significant thresholds of *p*-value and gene count for enrichment analysis were set to <0.05 and ≥2, respectively.

### 4.11. Cluster and Functional Annotation Analyses of PPI Networks

To extract a highly connected cluster from the above-mentioned constructed PPI networks, we investigated each statistically significant cluster by the MCODE [45] clustering algorithm. The clusters of highly connected nodes were identified at the following parameters: degree cut-off = 10, node score cut-off = 0.2, *K*-core = 2, and max depth up to 100. Bader and Hogue have described the principle and operating procedures of the MCODE clustering algorithm and have defined each parameter in detail [45]. Then, we investigated the functional pathways of each statistically significant cluster to identify the major functions of the clusters using the ClueGO [46] application in Cytoscape. KEGG was employed as a database for pathway analysis, and two parameters of the ClueGO application, the *p* value and kappa score, were set to 0.05 and 0.3, respectively. The gene pathway functional networks of each cluster were visualized by the ClueGO plug-in in the Cytoscape software.

### 4.12. Identification of Potential Markers among the DEGs of Whisker Follicles or of the Striatum

To identify the potential markers among the DEGs of each tissue, we constructed a Venn diagram from the genes identified by three procedures: correlation analysis, centrality analysis, and cluster analysis. The blue and orange color of the gene name represented genes that were down-regulated or up-regulated in each tissue, and the red color represented the up-regulated genes common for the two tissues. To investigate functional linkages between the two tissues, we performed KEGG pathway enrichment analysis on potential markers of the striatum (49 genes) and of the whisker follicle (129 genes) and visualized the gene pathway network using the ClueGO application in Cytoscape.

### 4.13. Data Availability

The sequencing data analyzed during the current study are available in the Sequence Read Archive (SRA) database (accession number PRJNA633547).

## Figures and Tables

**Figure 1 ijms-21-08856-f001:**
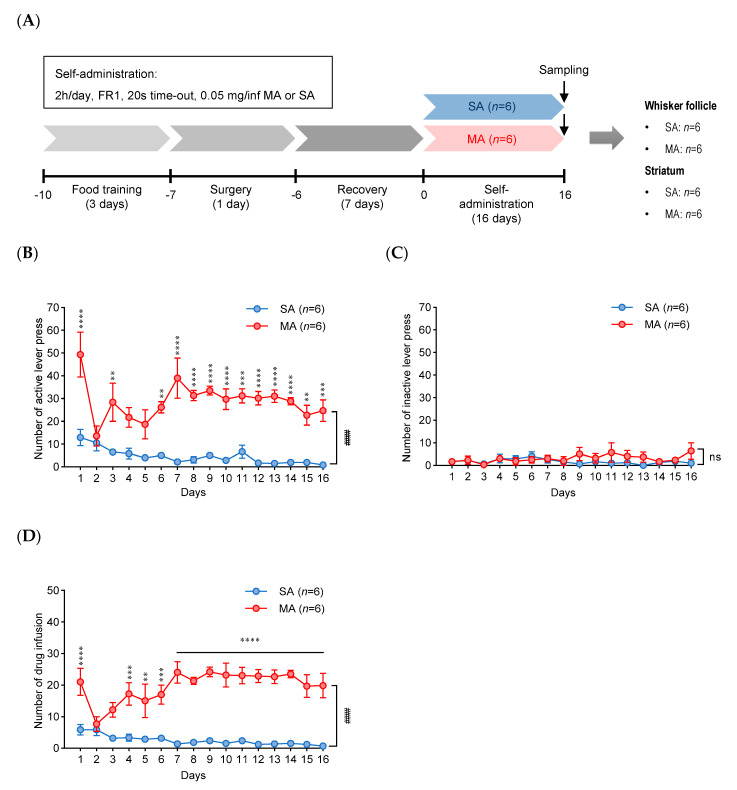
Methamphetamine (MA) self-administration. (**A**) The schematic diagram of the experimental timeline of saline (SA) or methamphetamine (MA) self-administration (*n* = 6 per group). (**B**,**C**) The numbers of active and inactive lever presses by SA or MA self-administration. (**D**) The number of infusions during MA self-administration at 0.05 mg/kg (2 h/day, fixed-rate 1 (FR1), 20 s time-out). Data were displayed as mean ± SEM (*n* = 6/group) and were subjected to two-way repeated measure ANOVA, followed by Bonferroni’s multiple comparison post hoc test: ** *p* < 0.01, *** *p* < 0.001, and **** *p* < 0.0001 on each day; #### *p* < 0.0001 between groups; *ns*, not significant.

**Figure 2 ijms-21-08856-f002:**
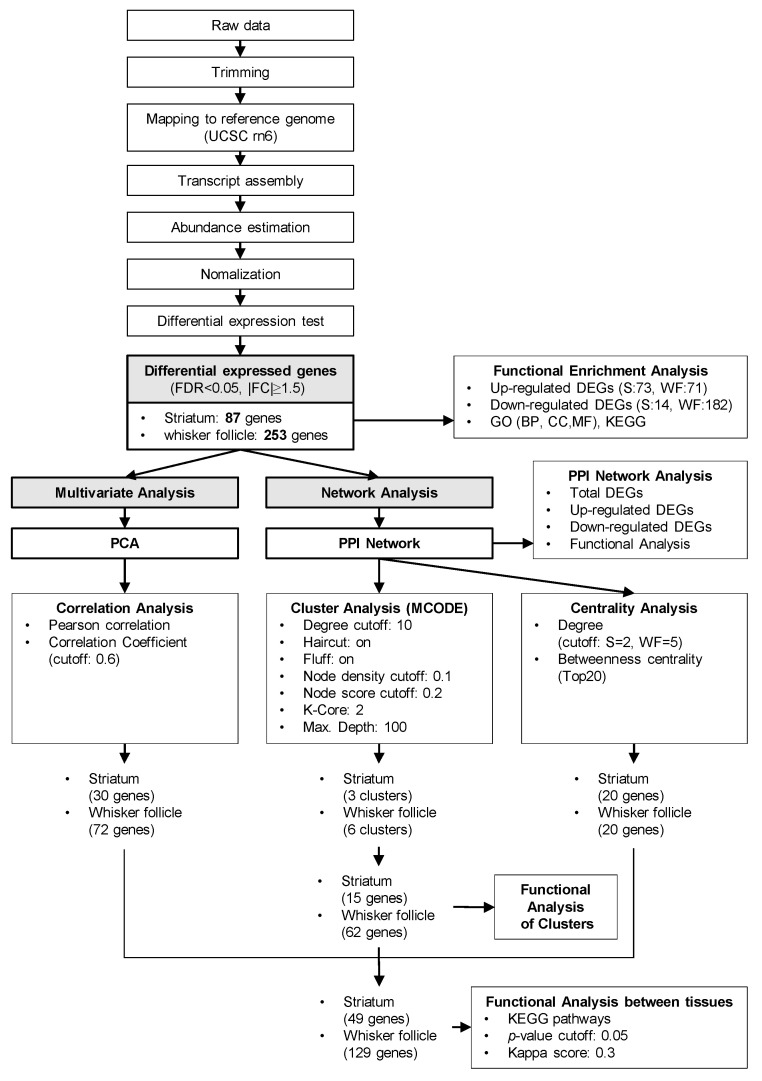
The workflow of computational analysis of the RNA sequencing (RNA-seq) data. A workflow diagram for the quality control and differential gene expression analysis of the RNA-seq data from the striatum or from the whisker follicle of MA self-administered rats.

**Figure 3 ijms-21-08856-f003:**
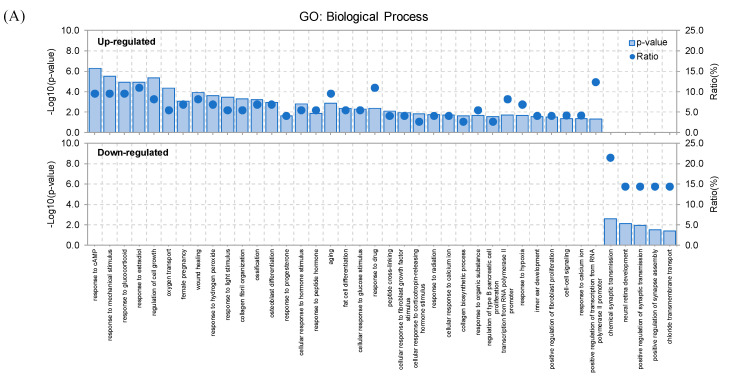
Functional enrichment analysis of up- and down-regulated differentially expressed genes (DEGs) in the striatum (**A**–**D**) and whisker follicle (**E**–**H**). Gene Ontology (GO) and Kyoto Encyclopedia of Genes and Genomes (KEGG) pathway were analyzed in up- and down-regulated differentially expressed genes (DEGs) using Database for Annotation, Visualization, and Integrated Discovery (DAVID) database. The horizontal axis represents various functional terms, including biological process (BC) (**A**,**E**), cellular component (CC) (**B**,**F**), molecular function (MF) (**C**,**G**), and KEGG pathway (**D**,**H**). The vertical axis represents the *p*-value (−Log10) and ratio (%) of the number genes enriched in GO and KEGG terms.

**Figure 4 ijms-21-08856-f004:**
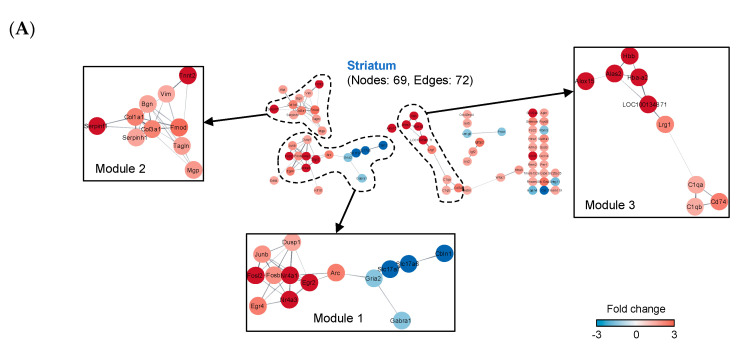
Construction of protein–protein interaction networks (PPI) from DEGs. (**A**) Construction of a PPI network in the striatum. (**B**) Construction of a PPI network in the whisker follicle. The putative PPI networks of DEGs were visualized via the Search Tool for the Retrieval of Interacting Genes/Proteins (STRING) application (confidence (score) cut-off = 0.4) in the Cytoscape software (v3.7.1). Red nodes denote up-regulated genes, and blue nodes indicate down-regulated genes. The intensity of each node color means the fold change. The top three modules of each tissue were identified by the MCODE plug-in in the Cytoscape software.

**Figure 5 ijms-21-08856-f005:**
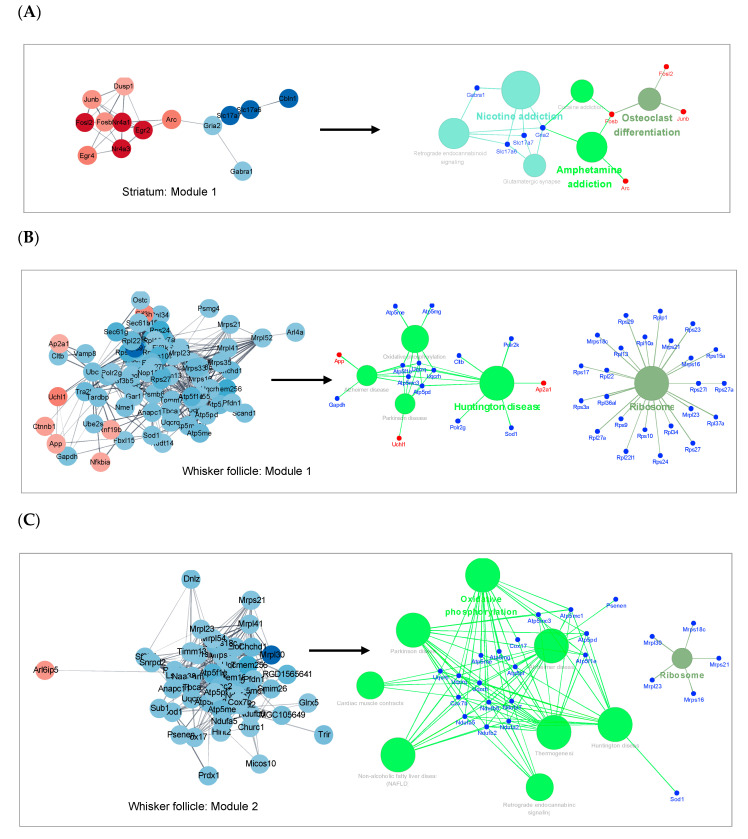
Cluster analysis of DEGs in whisker follicles or in the striatum. (**A**–**C**) Gene pathway networks that represent significantly enriched KEGG terms and related genes in designated modules. The sub-networks were analyzed and visualized using the ClueGO plug-in in the Cytoscape software.

**Figure 6 ijms-21-08856-f006:**
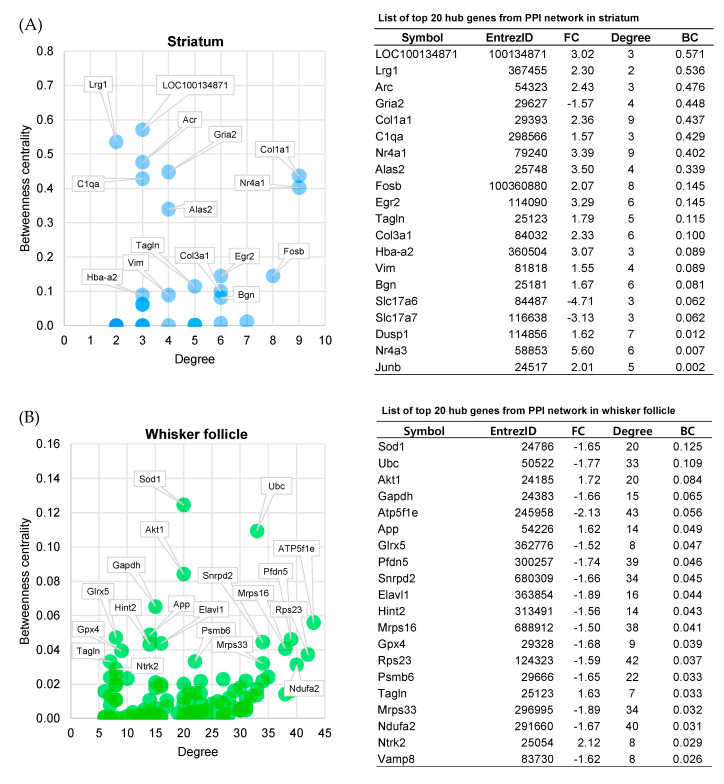
Centrality analysis and lists of the top 20 hub genes. (**A**,**B**) Betweenness centrality values and degrees of the entire PPI networks were obtained by means of the NetworkAnalyzer tool in the Cytoscape software (v3.7.1). Cut-offs of the degree for PPI networks in each tissue were set to 2 (striatum) and 5 (whisker follicle). Hub genes were arranged in the order of decreasing betweenness centrality values. A scatter plot of betweenness centrality versus the degree for nodes from PPI networks (left panel). The top 20 hub genes of PPI networks are listed in the lower panel. FC, fold change; BC, betweenness centrality.

**Figure 7 ijms-21-08856-f007:**
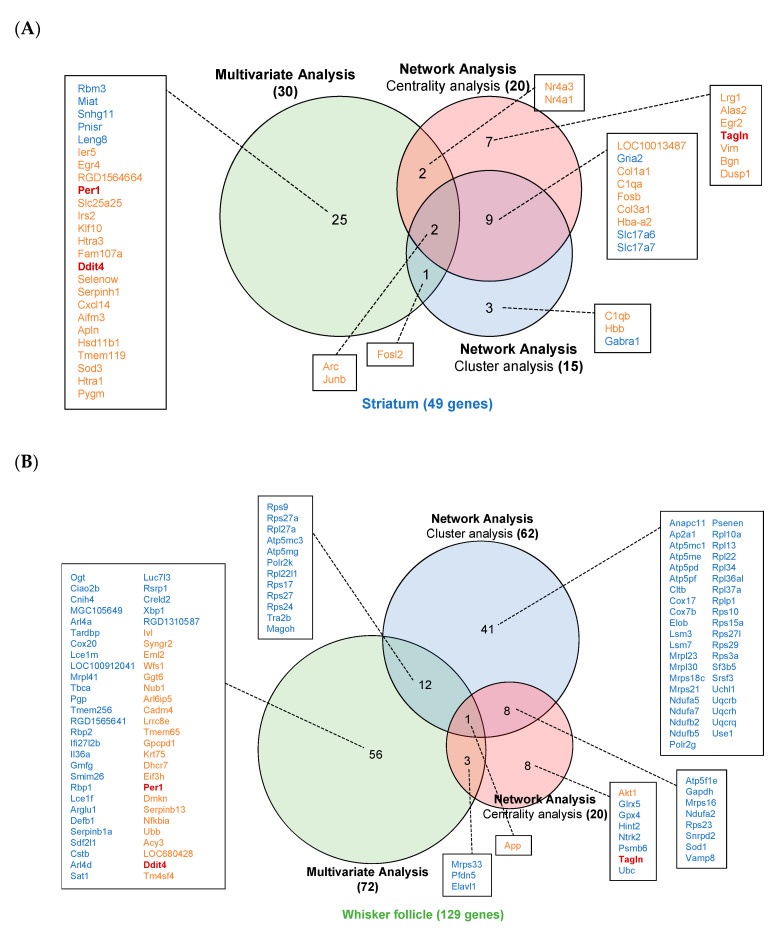
Potential diagnostic markers identified in the striatum (**A**) and in whisker follicles (**B**) of methamphetamine (MA) self-administered rats. The Venn diagram shows the potential markers identified by the multivariate and network analyses. Orange color: genes increased by MA, Blue color: genes decreased by MA, Red boldfaced: The genes identified both in the striatum and in whisker follicles.

**Figure 8 ijms-21-08856-f008:**
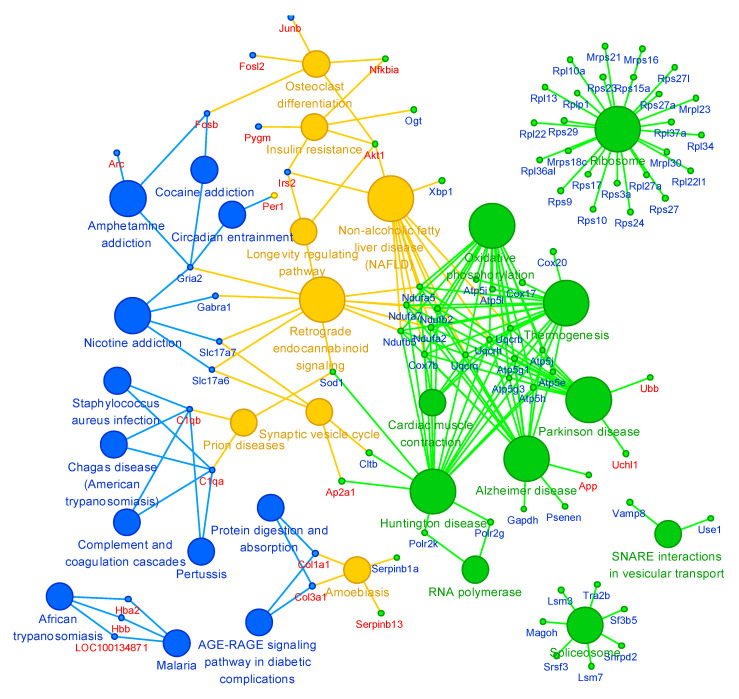
The interaction network between the striatum and the whisker follicle of methamphetamine self-administered rats. Blue color: A striatum-specific pathway (big circles) or genes (small circles), green color: a whisker follicle-specific pathway (big circles) or genes (small circles), and yellow color: the pathways or genes common for the two tissues.

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
