# Peer review of "Transcriptional Profiling of Whisker Follicles and of the Striatum in Methamphetamine Self-Administered Rats"

_ijms, 2020, doi:10.3390/ijms21228856_

Round 1
Reviewer 1 Report
This manuscript reported the transcriptional changes in whisker follicles and striatum after methamphetamine self-administration in rats. The authors performed RNAseq on whisker follicles and striatum from control and methamphetamine self-administrated rats, and identified differentially expressed genes and key pathways that are changed by methamphetamine treatment in whisker follicles and striatum. The authors also identified several genes and pathways which are altered in both tissues. Overall, the manuscript is well-written, and the overall data is clear.
- It has been reported by multiple papers that methamphetamine treatment produces long-lasting damage to striatal nerve terminals through dopamine and glutamate signaling. Did the authors detect the changes of these signaling/receptors in striatum? It would be more convincing if the authors can highlight these changes and show that the experiment is consistent with previous reports.
- Figure 1 B-D, p-value is needed to show the significance.
- Figure 3 A-B, PCA should be performed on gene expression profiles but not only DEG.
- Figure 6 A, labeling of Arc gene should be corrected (not Acr).
Reviewer 2 Report
Dear authors,
The current manuscript aims to identify important genes and pathways in Methamphetamine (MA) use disorder by analyzing global transcriptome from Whisker Follicles and Striatum following MA self-administration in rats. The findings are interesting, the study is well designed and executed with comprehensive analysis of transcriptome data. However, the manuscript needs to address following concerns:
Reviewer comments:
- The figure 1A and the method described in 4.2 are not in line. The food training was carried out post-surgery (4.2) or before surgery (Fig.1A)?
- Though authors have performed 2-way ANOVA (in Fig 1B-1D), failed to indicate the p values on significant data points. Highlight the significant data points in Fig 1B-1D with ‘*’ and indicate the p-values considered significant in fig. legends & methods section.
- Authors indicated in p.3; line-105 that the expression pattern is consistent between samples. But, the supportive figures are missing. Provide a figure indicating how the different groups and samples within each group were clustered (Hierarchical cluster or others)
- How the DEG’s were computed? Based on only FDR cut-off (<0.05) or Both FC>1.5 & FDR (<0.05)? In 4.6 section, authors used both the FDR & FC for DEG’s, but elsewhere in the manuscript mentioned only FDR as criteria for DEG’s (Fig.2 & p.3;line 99-101).
- If FC>1.5 were selected as indicated in 4.6 section, authors showed genes less that FC 1.5 in the list of DEGs (Table S1 & S2).
- The Fig.3 is more of qualitative and really do not provide much to the study conclusions rather than colorful-plot in the main manuscript. So, I suggest authors to move Fig.3 to supporting data file and bring the pathway analysis (whole or at least the important GO-MF & KEGG in C-D in Fig. S1&2) in to main manuscript. The results section 2.2 had ended abruptly just be stating that the authors did pathway analysis without mentioning any significant ones.
- The authors have published RNA-Seq analysis in Whisker-follicles followed by MA-self administration earlier. Is there any correlation/similarity in DEGS observed in this study? The reviewer recommends authors to compare the DEGS with previously published data and discuss in the section 2.2 or 3.
- Does the top hub 20 genes listed in Fig. 6 belong to same functional class? The section 2.6 looks like a detailed method rather than findings. Authors should try to extrapolate the hug genes to some major function or pathway to draw some conclusion out of the analysis carried out.
- The discussion can be improved and can considerate the finding sin whole rather than just focusing on three common genes identified (Per1, Ddit4, and Tagln)
- The transcriptome was carried-out after 16-days of MA self-administration. How the sampling time was optimized? Based on which parameters (elaborate on methods)? To get the real picture how MA-administration is affecting transcriptome, it should have been collected one sample on 8th day or so to see what genes are critical in early exposure to MA and leads to phenotype on day 16.
- After all, the findings were only computational and lacks the validation or verification of at least the key genes by qPCR or so. Considering the reputation of the journal, reviewer feels that the study lacks the scientific validation of the transcriptome data. As study aims to identify key genes or pathways for diagnosis or therapeutic purpose, these aspects were diluted in discussion section.
- Also, the reasons for concluding that whisker follicles can be alternative source of biomarkers for diagnosis of MA-disorder is not clear in the manuscript when there was not many genes were common in between both the tissues.
Minor comments:
- ‘four samples’ in p.3, line-106 should be corrected. Is it twenty four samples or four sampling groups? Clarify
- The reference for stratum collection is missing in p. 13, line-342. Only cites the paer for follicles collection.
- Indicate how many reads were collected for each sample and how many reads (%) mapped to ref genome (in results section).
- Is it FC>1.5 or log2-FC for DEGs identification?
- Which method was used to calculate the FDR (Benjamini-Hochberg/Bonferroni)?
- Cite the reference for ClueGO in 4.11 & 4.12.
- How many DEGs’ were common between follicles & striatum?
- The GO-MF fig for down-regulate genes is missing in Fig. S1. Please add it.
- Authors wrote that Arc, Junb, App genes were ‘filtered out’ giving a notion that these are not important ones. Use another term to other than ‘filtered out’
- Authors indicated in p.14, line-374 that a heatmap was generated by R-package. But, as such there was no heatmap was provided in the manuscript.
- List of abbreviations can be extended a bit.
Round 2
Reviewer 2 Report
Dear authors,
I really appreciate the efforts put in revising the manuscript and now it is in good shape. However, the current manuscript needs to incorporate few methodological details that were missing:
1) Please include 'Immunoblotting' & 'RT-PCR' methods in the Mat & Methods section. Also, provide the primer sequences used for RT-PCR validation & antibodies details (source, Cat.No).
2) Also indicate what house-keeping gene (ACTIN/GAPDH/ etc) was used for RT-PCR quantification/normalization.
3) Similarly, indicate which loading-control was used for western blotting and describe how the protein intensities were quantified in methods section. Incorporate the western blot images for loading control in Fig. S1 & Fig S5.
Best,
